# Harnessing CD8^+^CD28^−^ Regulatory T Cells as a Tool to Treat Autoimmune Disease

**DOI:** 10.3390/cells10112973

**Published:** 2021-11-01

**Authors:** Sabrina Ceeraz, Charlotte R. Thompson, Richard Beatson, Ernest H. Choy

**Affiliations:** 1Carisma Therapeutics, Philadelphia, PA 19104, USA; sabrina.delong@carismatx.com; 2Brighton and Sussex Medical School, University of Sussex, Brighton BN1 9RH, UK; c.thompson2@bsms.ac.uk; 3School of Cancer & Pharmaceutical Sciences, Faculty of Life Sciences and Medicine, King’s College London, London SE1 9RT, UK; richard.1.beatson@kcl.ac.uk; 4CREATE Centre, Division of Infection and Immunity, School of Medicine, Cardiff University, Cardiff CF14 4XN, UK

**Keywords:** CD8, Treg, autoimmunity, immunoregulation

## Abstract

T regulatory cell therapy presents a novel therapeutic strategy for patients with autoimmune diseases or who are undergoing transplantation. At present, the CD4^+^ Treg population has been extensively characterized, as a result of defined phenotypic and functional readouts. In this review article, we discuss the development and biology of CD8^+^ Tregs and their role in murine and human disease indications. A subset of CD8^+^ Tregs that lack the surface expression of CD28 (CD8^+^CD28^−^ Treg) has proved efficacious in preclinical models. CD8^+^CD28^−^ Tregs are present in healthy individuals, but their impaired functionality in disease renders them less effective in mediating immunosuppression. We primarily focus on harnessing CD8^+^ Treg cell therapy in the clinic to support current treatment for patients with autoimmune or inflammatory conditions.

## 1. Introduction

The immune system is a comprehensive network of multifaceted cells that exert an array of systemic and tissue-dependent functionalities. Pro- and anti-inflammatory responses are required to initiate an effector response against infections or cancers while maintaining tolerance to self-antigens and driving tissue repair. Several mechanisms play a key role in maintaining immune tolerance such as immunosuppressive cytokines, a high threshold for T cell activation and T regulatory cells (Tregs).

In the 1970s, Gershon and Kondo showed that T cells expressing Lyt2 (CD8α) could mediate suppressor activity, with others subsequently confirming the suppression of antigen specific responses [1,2]. However, a lack of tools and markers needed to identify suppressor function, compounded by the strong association of CD8^+^ T cells with cytolytic (CTL) function, led to dormancy in the field of CD8^+^ suppressor cells. In the mid-1990s, the emergence of CD4^+^ Tregs expressing the robust marker forkhead/winged helix transcription factor (Foxp3) [3,4,5] and association between Foxp3 gene mutations and fatal X-linked syndrome [6,7] revived interest in suppressor/regulatory cells. Our understanding of CD4^+^ Tregs has evolved to include a relatively small set of phenotypic and functional markers such as the expression of Foxp3, CD127^−^/low and helios and inhibitory cytokine production: IL-10, IL-35 and transforming growth factor beta (TGF-β1). In contrast, a vast array of markers and functions have been reported for CD8^+^ suppressor cells (now referred to as CD8^+^ Tregs) in mouse and human studies. The aim of this review is to discuss the renewed interest in the CD8^+^CD28^−^ Treg subset, as a result of improved phenotypic markers, preclinical studies and detection in patients with autoimmune and inflammatory conditions. 

## 2. Origin of CD8^+^ Tregs

To date there have been no reports of CD8^+^ natural Tregs. Like CD4^+^ Tregs, continuous antigen stimulation of CD8^+^CD25^−^ T cells results in the generation of CD8^+^Foxp3^+^ Tregs in response to Staphylococcus enterotoxin B [8]. In vitro, co-culture of CD8^+^CD28^−^ T cells with monocytes is shown to generate suppressor cells that require IL-10 to inhibit CTL function and T cell proliferation [9]. Similarly, others have shown in vitro studies that CD3-stimulated co-cultures of bone-marrow-derived stromal cells with CD8^+^ T cells result in human CD8^+^ Tregs that express CD25, CD28 and Foxp3 [10]. 

CD8^+^ Tregs develop naturally or can be induced by T cell receptor (TCR) or non-TCR signals [11], mediating suppression by cell contact, cytokine and chemokine production, MHC class 1 restriction, cytotoxicity or indoleamine 2, 3-dioxygenase (IDO) production [11].

In pregnancy, human placental trophoblasts can activate CD8^+^ Tregs [12] and, in the eye, ocular pigment epithelial cells can covert CD8^+^ T cells to CD8^+^CD25^+^Foxp3^+^ Tregs [13]. This demonstrates that CD8^+^ Tregs can mediate suppression in immune privilege sites.

CD28 loss is associated with chronic antigen stimulation [14], and, pertinent to chronic inflammatory disease, TNF-α has been highlighted in driving down expression in an autocrine or paracrine fashion [15]. As such, CD28 loss is considered a marker for both the generation of effector memory and terminal effector memory CD45RA^+^ (TEMRA) subsets, and CD8^+^ Tregs.

## 3. CD8^+^ Treg Populations in Mice and Humans

Similarities between CD4^+^CD25^+^ Tregs and CD8^+^CD122^+^ Tregs include the ability to suppress T cell activity and contribute to T cell homeostasis [16]. CD8^+^CD122^+^ Tregs are naturally occurring and resemble a memory phenotype. In mice, IL-10 producing CD8^+^CD122^+^ Tregs mediate the suppression of activated T cells via MHC class I αβ TCR signaling rather than recognition by MHC class II (I-A) and MHC class Ib molecule Qa-1 [17]. 

The negative checkpoint regulator (NCR) programmed death-1 (PD-1) receptor is critical for CD8^+^ T cell homeostasis, exhaustion and response to viral infections. In a murine cardiac transplantation model, inducible co-stimulator (ICOS)-B7h blockade induces alloantigen CD8^+^CD122^+^ Tregs expressing PD-1, which contribute to prolonging allograft survival via a Th2 response [18]. In addition, CD8^+^CD122^+^ Treg function has been reported in murine anti-tumor responses against the murine mouse lymphoblastic lymphoma cell line (EL4) and in B16-OVA and CT26 [19,20]. In humans, the equivalent of CD8^+^CD122^+^ Tregs are CD8^+^CXCR3^+^ T cells, which are distinct from central memory cells [21]. In humans and mice, CD8^+^Foxp3^+^ Tregs have been reported. In humans, CD8^+^Foxp3^+^ Tregs are detected in some patients with rheumatoid arthritis (RA) [22], systemic lupus erythematosus (SLE) [23], Epstein-Barr virus (EBV) infections [24], patients with prostate cancer [25] and allogeneic stem cell transplantation [26]. In mice CD8^+^ Tregs expressing Foxp3 are reported in lupus-prone mice [27], allogeneic stem cell [26] and bone marrow transplantation [28].

Recent studies have identified suppressive CD8^+^CD103^+^ Tregs that express CD39 that inhibit lupus nephritis in a graft-versus-host disease (GvHD) model [27] or alloprimed CD8^+^ T cells expressing CXCR5 suppress post-transplant alloantibody production [29]. CD8^+^ T cells that lack surface expression of CD28 (CD8^+^CD28^−^ Treg) have been widely documented in autoimmune diseases and inflammatory conditions [30,31,32]. We have previously shown that, in RA patients, CD8^+^CD28^−^ Tregs are dysfunctional due to defective suppressor function and/or the reduced responsiveness of target cells to suppression, which can be corrected by anti-TNF therapy [33]. CD8^+^ Treg markers in mice and humans are summarized in Figure 1.

## 4. Mechanisms of Suppression Mediated by CD8^+^ Tregs

The ability of CD8^+^ Tregs to suppress inflammation is partly mediated by targeting autoreactive or alloantigens, as well as conventional CD4^+^ T cells. Several studies have demonstrated that CD8^+^ Tregs suppress activated antigen-specific CD4^+^ T cells expressing Qa-1 [34]. As Qa-1 binds self-peptides that activate natural killer (NK) receptors on CD8^+^ T cells, Qa-1 may play a critical role in CD8^+^ Treg homeostasis [34].

In addition to suppressing T cell proliferation, there is now increasing evidence that CD8^+^ Tregs impact the B cell compartment. One mechanism is the inhibition of T follicular helper (Tfh) cells that highly express Qa-1 resulting in a reduction in germinal center (GC) formation, antibody-switching and maturation [35]. In ApoE mice crossed with Qa-1° mice or Qa-1 D227K mice, defective CD8^+^ Treg function accelerates atherosclerosis due to enhanced Tfh and GC function [36].

Alternative mechanisms reported for CD8^+^ Treg suppression include apoptosis [37] and the expression of TNFR2 or PD-L1 [38]. A recent study has shown that the transcription factor (TF) STAT4 may play an intrinsic role in CD8^+^ Tregs. This is demonstrated in vivo by the adoptive transfer of Stat4-deficient CD8^+^ Tregs into C57BL/6 (CD45.1^+^) mice immunized with KLH/CFA, reducing Tfh cells and GC B cell formation. Similarly, in a murine atherosclerosis model, the adoptive transfer of *Stat4*^−/−^*Ldlr*^−/−^ CD8^+^CD122^+^ Tregs into *Ldlr*^−/−^ recipients suppressed plaque formation, GC B cells, Tfh cells and Ig formation [39]. Under Treg skewing conditions, the polarization of splenic *Stat4*^−/−^ CD8^+^ T cells to CD8^+^ Tregs is increased in vitro, and *Stat4*^−/−^ macrophages inhibit Tfh cells but enable CD8^+^ Treg differentiation. In contrast, *Stat5* deficiency in the CD8^+^ T cell compartment is linked to impaired B cell tolerance due to an increase in CD8^+^ Tfh, resulting in increased autoantibody production [40]. These findings suggest TFs can serve as tool in studying CD8^+^ Treg populations and their differentiation [39,40].

## 5. CD8^+^CD28^−^ Tregs

CD8^+^CD28^−^ Treg suppressor function has been reported in normal healthy individuals [41] and patients with cancer [42], those who are undergoing transplantation (rejection free recipients) [43,44,45] or who have gastrointestinal [46] or autoimmune [31] conditions. CD8^+^CD28^−^ Treg generation by either allogeneic stimulator cells [47], antigen-presenting cells (APC) [48] or xenogeneic stimulator cells [47] has complicated the understanding of how these cells mediate suppression.

Overlapping markers between CD8^+^CD28^−^ Tregs and CD4^+^CD25^+^ Tregs include the expression of Foxp3, CD25 and members of the TNF receptor family (e.g., 4-1BB) [49]. A unique feature mediated by CD8^+^CD28^−^ Tregs is their ability to render APCs tolerogenic by reducing CD86 gene transcription [50] while also upregulating inhibitory receptors, such as immunoglobulin-like transcript (ILT) 3 (ILT3) and ILT4 [51].

To date, no studies have robustly characterized the molecular pathways involved in the generation and activation of CD8^+^CD28^−^ Tregs; however, mixed lymphocyte culture (MLC) stimulation in the presence of IL-2, IL-7 and IL-15 has shown promise in generating and expanding these cells [52]. These data in combination with those described above are suggestive of JAK-STAT involvement.

## 6. CD8^+^CD28^−^ Tregs in Autoimmunity

### 6.1. Multiple Sclerosis

In experimental autoimmune encephalomyelitis (EAE), the murine model of multiple sclerosis (MS), the adoptive transfer of CD8^+^CD28^low^ T cells from wild-type (WT) mice into recipients lacking CD8 significantly suppress disease severity, unlike CD8^+^CD28^high^ cells. This finding was supported in vitro where CD8^+^CD28^low^ T cells reduced interferon-gamma (IFN-γ) production from myelin oligodendrocyte glycoprotein (MOG) (MS autoantigen)-specific CD4^+^ T cells and rendered APCs tolerogenic by downregulating CD40, CD80 and CD86 [53]. In EAE mice treated with a trichosanthin-derived peptide, CD8^+^CD28^low^ Tregs producing IL-10 also reduced the clinical score [54]. In MS and type 1 diabetes (T1D) patients, the frequency of CD8^+^CD28^−^ Tregs is reduced compared with healthy individuals [32].

T cell vaccination models further demonstrate that CD8^+^ T cell suppressor function exerts a profound effect on controlling the onset of autoimmunity. For example, the adoptive transfer of T cells, in particular CD8^+^ from vaccinated mice, prevents overt EAE, owing to suppressor properties in the CD8^+^ T cell compartment [55]. This is confirmed in MS patients, in whom CD8^+^ Tregs inhibit proliferation and lyse vaccine CD4^+^ T cells clones in vitro [56]. A subsequent study showed that, in CD4^+^ T-cell-vaccinated mice, Vβ8-specific Qa-1-restricted CD8^+^ T cells are induced alongside activated CD4^+^ Vβ8^+^T cells and lyse Qa-1 and murine TCR Vβ8^+^ cells, with a similar finding in mice vaccinated with autoantigen myelin basic protein (MBP) [57].

### 6.2. Rheumatoid Arthritis (RA)

In 2005, Davila and colleagues presented a study exploring a CD8^+^ T cell-based immunotherapy strategy in RA by adoptively transferring CD8^+^CD28^−^CD56^+^ T cell clones from RA synovium in NOD-SCID mice engrafted with synovial tissue that suppressed T cell infiltration, IFN-γ and TNF-α levels [58].

In RA patients, impaired Treg function highlights the importance of harnessing Tregs as a therapeutic tool. We have shown that CD8^+^CD28^−^ Tregs are significantly increased numerically in the peripheral blood (PB) of RA patients treated with methotrexate (RA(MTX)) but exert reduced suppressor function compared to healthy individuals [33]. In patients treated with TNF inhibitors (TNFi), suppressive potential was partially restored demonstrating the impact of therapy on Treg function. Phenotypic analysis of RA(MTX) CD8^+^CD28^−^ Tregs revealed that the cells expressed low levels of ICOS and PD-1 compared with healthy individuals. Although the cells present in the synovial fluid (SF) and PB express IL-10 and mediate suppression partially by TGF-β1, deficits in both the CD8^+^CD28^−^ Treg, and the responsiveness of the target cells to suppression, hinder their full suppressive potential [33]. The increase we observed in the PB of RA patients was confirmed by Thompson and colleagues, who also showed that the expansion of CD8^+^CD28^−^ Treg occurs early in the disease, correlated with disease duration and was associated with previous cytomegalovirus (CMV) infection [59].

In addition to CD8^+^CD28^−^ Tregs, an anti-CD3 monoclonal antibody (mAb) has been shown to activate peripheral blood mononuclear cells (PBMC) from RA patients to induce CD8^+^Foxp3^+^ Tregs that inhibit IL-17 production [22]. In RA, the induction of these cells was regulated by p38 phosphorylation in co-cultures of naïve CD8^+^ T cells in response to membrane-bound TNF-α and CD86 expressed on monocytes [22]. In a recent report, the assessment of SF of a subset of arthritis patients reduced TCR-ζ in CD8^+^CD28^null^ expansion, which correlates with increased inflammation and neutrophil migration. Similarly, low levels of ZAP-70 in expanded CD8^+^CD28^low^ was associated with Disease Activity Score 28 (DAS28). The predominance of downregulating TCR-ζ in SF CD8^+^CD28^null^ cells in patients with seronegative arthritides suggests that TCR molecules in CD8^+^CD28^−^ cells may serve as a biomarker in RA [60].

### 6.3. Systemic Lupus Erythematosus (SLE)

In the murine lupus strain, New Zealand Black/New Zealand White F1, female mice treated with an artificial synthetic peptide (pConsensus) containing MHC class I and II determinants in the VH of an anti-DNA Ig, displayed increased CD8^+^ Tregs (CD28^+^ and CD28^−^). These cells suppressed anti-DNA Ab production [61] and T cell proliferation [62] in a Foxp3- and TGF-β1-dependent manner [61]. Furthermore, CD8^+^ Treg function was impaired in mice treated with pConsensus peptide in the presence of PD-1 blockade, indicating that NCRs were important for their function [63].

Two human studies have shown conflicting observations for CD8^+^CD28^−^ Treg detection in SLE patients. The first showed that CD8^+^CD28^−^ T cells are reduced in the PB of SLE patients compared to healthy or disease controls; this observation together with low levels of IL-10 and *TGFB1* mRNA levels may account for defective suppressor function in disease [31]. In contrast, a second study reported a significant increase in the absolute number and percentage of CD8^−^CD28^−^ T cells in SLE patients compared to healthy individuals, which positively correlated with SLEDAI (SLE disease activity index). Furthemore, in patients with active lupus nephritis, a positive correlation between CD8^+^CD28^−^ T cell number expressing low levels of Foxp3 and disease activity was noted [64]. Hence, in SLE patients, the potent pro-inflammatory environment may hinder suppressor function.

### 6.4. Diabetes

In type 1 diabetic patients, treatment with an anti-CD3 mAb has been shown to induce CD8^+^CD25^+^ Tregs, which express cytotoxic T-lymphocyte-associated protein (CTLA-4) and Foxp3 to facilitate the inhibition of CD4^+^ T cells in response to therapy [65]. A recent study addressing how CD8^+^ Tregs may mediate suppression in diabetes reported that intestinal levels of *Ruminococcus* positively correlated with the number of CD8^+^ Tregs in the PB [66].

### 6.5. Systemic Sclerosis (SSc)

In patients with systemic sclerosis (SSc), CD8^+^CD28^−^ T cells are increased in the blood and skin with suppressor and cytotoxic function respectively [67,68]. In the skin, resident CD8^+^CD28^−^ T cells are highly profibrotic, suggesting that suppressor potential is dependent on the microenvironment. In vitro assessment of non-antigen specific CD8^+^CD28^−^ Tregs from SSc patients further confirmed their impaired ability to suppress antigen-specific CD4^+^ T cell proliferation [9].

## 7. CD8^+^CD28^−^ Tregs in Gastrointestinal Disease

CD8^+^ Tregs play a critical role in mucosal tolerance. Adoptive transfer of naïve CD4^+^CD45RB^high^ T cells (colitogenic) with splenic CD8^+^CD28^low^ T cells into syngeneic immunodeficient Recombinase Activating Gene-2 (*RAG-2*) mutant mice, prevent IBD onset via IL-10 production. Notably, the authors showed that CD8^+^CD28^low^ T cells isolated from the intestinal epithelium and lamina propria (LP) also had a similar protective effect mediated by IL-10 production and the responsiveness of colitogenic T cells to TGF-β1 [69].

In a model of experimental colitis, the autoimmune regulator (AIRE)-deficient CD8^+^CD28^low^ Treg suppressor function was impaired compared to WT [70]. Furthermore, in mice intranasally injected with ovalbumin (OVA) encased in oligomannose-coated liposomes, a model of food allergy, induced CD8^+^CD28^low^ Tregs and CD4^+^CD25^+^Foxp3^+^ Tregs in the mesenteric lymph node reduced allergic diarrhea [71]. In a rat model of trinitrobenzenesulfonic acid-induced colitis, treatment with mesalazine, a therapy for ulcerative colitis (UC), increased the ratio of CD8^+^CD28^+^/CD8^+^CD28^−^ T cells in the blood. In the colon, a reduction in CD8^+^CD28^−^ T cells contributed to disease progression [72]. In human studies, the CD8^+^CD28^+^/CD8^+^CD28^−^ T cell ratio of <1.03 in the PB acts as a prognostic tool in Crohn’s disease (CD) patients progressing to an active disease stage [73]. In UC patients, a shift in the cell balance in favor of CD8^+^CD28^+^ T cells in the PB and colon tissue is desirable [30].

In patients with inflammatory bowel disease (IBD), the frequency of LP CD8^+^ Tregs expressing TCRVβ5.1 are reduced compared with non-IBD inflammatory and normal LP. In vitro, LP CD8^+^ T cells (CD28^+^ and CD28^−^) from patients with UC or CD fail to suppress Ig production by PBMC treated with pokeweed mitogen compared with normal LP [74].

Human intestinal epithelial cells (IECs) can induce the proliferation of CD8^+^CD28^−^, CD8^+^CD28^+^ and CD8^+^CD28^−^CD101^+^CD103^+^ T cells in the presence of gp180, a 180-kDa epithelial membrane glycoprotein. In the presence of IL-7 and IL-15, IEC-stimulated CD8^+^ T cells are rendered suppressive, especially when assessing the CD101^+^CD103^+^ population. The authors showed that suppression mediated by CD8^+^CD28^−^CD101^+^CD103^+^ T cells required direct contact with gp180 [46] recognized by CD1d. Hence, a possible explanation for impaired regulatory function may be low or altered gp180 expression by IEC in UC, CD or IBD patients [75].

## 8. Viral Infections

CD8^+^ Tregs have been documented in numerous viral infections. In individuals infected with hepatitis B virus (HBV), an increase in CD8^+^CD28^−^ T cells in the chronic phase of infection negatively correlates with alanine aminotransaminase (ALT) and aspartate aminotransferase (AST) levels, suggesting improved liver health [76]. In human immunodeficiency virus (HIV) infected patients, an expansion in CD8^+^CD28^−^ T cells has been reported in the blood and lung [77]. Similarly, these cells are additionally expanded in the blood of RA patients with Epstein-Barr Virus (EBV) infection [78].

## 9. Transplantation

There is now increasing evidence that the interplay between co-inhibitory and co-stimulatory molecules plays a role in CD8^+^CD28^−^ Treg function. In transplantation studies, CD8^+^ T cells from B6 recipients of BALB/c heart allografts undergoing anti-ICOS Ab treatment prolong allograft survival, unlike CD8^+^ T cells isolated from control mice [18].

## 10. Cancer

Increasing evidence points towards CD4^+^CD25^+^Foxp3^+^ Tregs having a pathogenic role in cancer development and metastasis, primarily through inhibiting host cancer-associated-antigen adaptive immunity [79,80]. As the numerically dominant Treg population in the tumor microenvironment (TME), CD4^+^CD25^+^Foxp3^+^ cells have been extensively studied in cancers, unlike CD8^+^CD28^−^ Tregs. However, the few studies there are suggest similarities between the two populations. For example, an increase in CD8^+^CD28^−^ Tregs has been seen to correlate with poor prognosis in metastatic cancer patients receiving T-lymphocyte immunotherapy following chemotherapy [81], and the expression of CD39 on CD8^+^ Tregs has been linked to the evasion of the antitumor immune response [82].

Inhibiting the impact of CD4^+^CD25^+^Foxp3^+^ Tregs through checkpoint blockade (CTLA4 and PD-L1;PD-1) has led to significant therapeutic breakthroughs in solid cancers [83]. Work is ongoing to try to deplete these cells, e.g., through CTLA-4 targeting, or inhibit their generation, e.g., by inhibiting TGF-β1 signaling [80,84]. These concepts can equally be applied to CD8^+^CD28^−^ Tregs; indeed, with CTLA-4 being reported on these cells [65], some of these current therapies may already be impacting this population.

The downregulation/loss of CD28 expression on the surface of CD8^+^ T cells has been shown to coincide with normal aging [14]. Senescent CD8^+^CD28^−^ T cells are oligoclonal and express NK-associated molecules CD56, CD57 and CD94 [85,86]. In cancer, wherein senescent cells impact the immune response, further studies are warranted to examine the anergic, activation and exhausted phenotype compared to CD8^+^CD28^−^ Tregs.

## 11. Adoptive Cell Therapy

Cell therapies such as engineered T cells expressing a specific chimeric antigen receptor (CAR) are efficacious as a non-solid cancer immunotherapy. A recent study wherein HLA-A*02 antigen-specific antibody (A2-CAR) human CD8^+^ Tregs reduced mismatched HLA-A*02 skin graft rejection and prevented xenogeneic GvHD in NSG mice [87] presents a novel platform for CD8^+^Treg therapy.

## 12. Concluding Remarks

In conclusion, this review highlights the importance of CD8^+^CD28^−^ Tregs in facilitating immune tolerance in inflammatory conditions. CD8^+^CD28^−^ Tregs display functional plasticity in accordance with their microenvironment (e.g., elevated, but defective, in active RA but protective in IBD), suggesting that the targeting or use of these cells will be highly specific for each disease indication. Indeed further knowledge regarding the factors that govern functionality may lead to therapies that modulate the impact of these cells in situ. Further to this, the field urgently requires consensus with regards to markers that are indicative of regulatory function, rather than the umbrella term of CD8^+^CD28^−^, which, as discussed, includes well-established inflammatory populations.

If a *bona fide* functional marker is found, not only could it be used to potentially deplete these cells in pathologies such as cancers, but also, pending robust large-scale expansion methodologies, it could be used to establish an anti-inflammatory adoptive cellular therapeutic for use in patients in whom current treatment is ineffective. 

## Figures and Tables

**Figure 1 cells-10-02973-f001:**
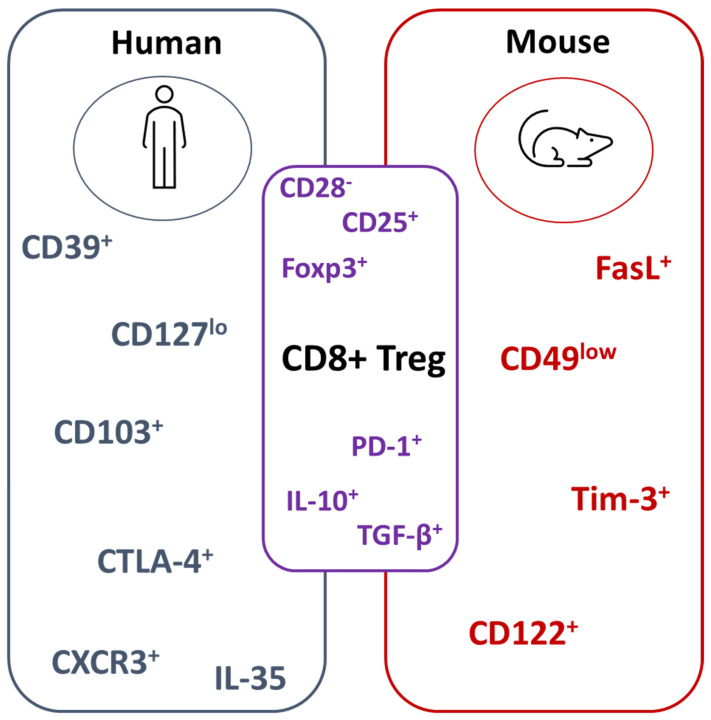
Summary of CD8^+^ Treg markers in mice and humans. CD8^+^ Treg phenotypic and functional markers differ between human and mouse studies. Well-characterized molecules between both species include the absence of CD28 and the expression of Foxp3. For functional readouts, IL-10 and TGF-β1 has been reported. Cytotoxic T-lymphocyte-associated protein 4 (CTLA-4); programmed death-1 (PD-1); T-cell immunoglobulin and mucin domain 3 (Tim-3); transforming growth factor beta (TGF-β).

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
