# Peer review of "Harnessing CD8+CD28− Regulatory T Cells as a Tool to Treat Autoimmune Disease"

_cells, 2021, doi:10.3390/cells10112973_

Round 1
Reviewer 1 Report
Regulatory T cells represent a novel therapeutic strategy for patients with autoimmune diseases or undergoing transplantation. As the authors stated, while the CD4+ Treg population has been extensively characterized less informations are available on biology of CD8+Tregs, and their role in murine and human disease indications.
In this review article the authors focused on a subset of CD8+ Tregs which lack surface expression of CD28 (CD8+CD28-Treg) and discuss available data on their impaired functionality in disease rendering them less effective in mediating immunosuppression. They primarily focus on harnessing CD8+Treg cell therapy in the clinic to support current treatment for patients with autoimmune or inflammatory conditions.
The manuscript is of interest and, in general, well organized.
However, in my opinion the authors should better introduce the topic of immunoregulatory cell populations to focus the specific role of CD8+CD28-Treg within the immune microenvironment and interaction with other immune cell populations.
-Among autoimmune diseases, I would suggest to recall the observation of a potential implication of CD8+ Treg in systemic sclerosis as previously reported (Non-antigen specific CD8T suppressor lymphocytes in diseases characterized by chronic immune responses and inflammation. Ann NY Acad Sci 2005;1050:115–23).
-Another topic worth to mention is the role of regulatory T cells in viral diseases. Previous findings demonstrated that CD28+CD8- and CD4+CD25high regulatory T cells might exert distinct effect on modulating antiviral immune responses and mitigate immunomediated liver damage in different phases of HBV infection, which represent potential prognostic markers and therapeutic targets for HBV-infected patients based on further exploration of detailed mechanism (Discrepant Clinical Significance of CD28+CD8- and CD4+CD25high Regulatory T Cells During the Progression of Hepatitis B Virus Infection. Viral Immunol. 2018;31(8):548-558).
-Role of CD8+ Treg in cancer: This is a topic of current major interest. The manuscript could be improved by adding a paragraph discussing the crucial role of some immune cell populations in the tumor microenvironment as recently well described in a comprehensive review addressing the role of CD4+CD25+ Treg in hepatocellular carcinoma (Hepatocellular carcinoma in viral and autoimmune liver diseases: Role of CD4+ CD25+ Foxp3+ regulatory T cells in the immune microenvironment. World J Gastroenterol. 2021;27(22):2994-3009). I would suggest to add a paragraph discussing the potential role of CD8+ Treg as target of new emerging immunotherapy.
-The authors should also discuss the questions that remain unanswered related to the expression of specific surface markers of CD8+CD28− Tregs , the characterization of molecular pathways involved in generation and activation, the reason for their anergic status in vitro and the possible existence of modulators.
Author Response
Dear Reviewer,
Many thanks for your time in appraising the manuscript. Your comments have greatly improved its scope and depth. We are very grateful for the care and consideration you have given the piece.
Please find below our point-by-point answers to your comments.
Best wishes.
Regulatory T cells represent a novel therapeutic strategy for patients with autoimmune diseases or undergoing transplantation. As the authors stated, while the CD4+ Treg population has been extensively characterized less informations are available on biology of CD8+Tregs, and their role in murine and human disease indications.
In this review article the authors focused on a subset of CD8+ Tregs which lack surface expression of CD28 (CD8+CD28-Treg) and discuss available data on their impaired functionality in disease rendering them less effective in mediating immunosuppression. They primarily focus on harnessing CD8+Treg cell therapy in the clinic to support current treatment for patients with autoimmune or inflammatory conditions.
The manuscript is of interest and, in general, well organized.
We very much appreciate the reviewer’s comments, and are particularly grateful that the reviewer also has an appreciation of the need for this type of review.
However, in my opinion the authors should better introduce the topic of immunoregulatory cell populations to focus the specific role of CD8+CD28-Treg within the immune microenvironment and interaction with other immune cell populations.
-Among autoimmune diseases, I would suggest to recall the observation of a potential implication of CD8+ Treg in systemic sclerosis as previously reported (Non-antigen specific CD8T suppressor lymphocytes in diseases characterized by chronic immune responses and inflammation. Ann NY Acad Sci 2005;1050:115–23).
Many thanks for the suggestion. We have adjusted the text in the SSc section to include the following.
In vitro assessment of non-antigen specific CD8+CD28-Tregs from SSc patients further confirmed their impaired ability to suppress antigen specific CD4+T cell proliferation [9].
-Another topic worth to mention is the role of regulatory T cells in viral diseases. Previous findings demonstrated that CD28+CD8- and CD4+CD25high regulatory T cells might exert distinct effect on modulating antiviral immune responses and mitigate immunomediated liver damage in different phases of HBV infection, which represent potential prognostic markers and therapeutic targets for HBV-infected patients based on further exploration of detailed mechanism (Discrepant Clinical Significance of CD28+CD8- and CD4+CD25high Regulatory T Cells During the Progression of Hepatitis B Virus Infection. Viral Immunol. 2018;31(8):548-558).
Many thanks for the suggestion; an excellent reference. We have adjusted the text to include it.
CD8+Tregs have been documented in numerous viral infections. In individual infected with hepatitis B virus (HBV), an increase in CD8+CD28-T cells in the chronic phase of infection negatively correlates with alanine aminotransaminase (ALT) and aspartate aminotransferase (AST) (indicators of disease) [76]. In human immunodeficiency virus (HIV) patients, an expansion in CD8+CD28- T cells is reported in the blood and lung [77]. Similarly, these cells are expanded in the blood of RA patients with EBV infection [78].
-Role of CD8+ Treg in cancer: This is a topic of current major interest. The manuscript could be improved by adding a paragraph discussing the crucial role of some immune cell populations in the tumor microenvironment as recently well described in a comprehensive review addressing the role of CD4+CD25+ Treg in hepatocellular carcinoma (Hepatocellular carcinoma in viral and autoimmune liver diseases: Role of CD4+ CD25+ Foxp3+ regulatory T cells in the immune microenvironment. World J Gastroenterol. 2021;27(22):2994-3009). I would suggest to add a paragraph discussing the potential role of CD8+ Treg as target of new emerging immunotherapy.
Many thanks. We very much agree. We have added the reference and have taken the opportunity to enhance the Cancer and discussion sections as detailed below.
In addition to the role of CD8+CD28-Tregs in autoimmune and inflammatory conditions, an increase in CD8+CD28-Tregs correlates with poor prognosis in metastatic cancer patients receiving T-lymphocyte immunotherapy following chemotherapy [79]. Furthermore, expression of CD39 on CD8+Tregs has been linked to evasion of the antitumor immune response [80].
In conclusion, this review highlights the importance of CD8+CD28− Tregs in facilitating immune tolerance in inflammatory conditions. CD8+CD28-Tregs display functional heterogeneity according to their microenvironment (e.g., elevated, but defective, in active RA but protective in IBD), suggesting that the targeting or use of these cells will be highly specific for each disease indication. Indeed further knowledge regarding the factors that govern functionality may lead to therapies that modulate the impact of these cells in situ. Further to this, the field urgently requires consensus with regards to markers which are indicative of regulatory function, rather than the umbrella term of CD8+CD28−, which, as discussed, includes well-established inflammatory populations. If a bonafide functional marker is found not only can it be used to potentially deplete these cells in pathologies such as cancers, but also, pending robust expansion methodologies, it could be used to establish an anti-inflammatory adoptive cellular therapeutic for use in patients where current treatment is ineffective.
-The authors should also discuss the questions that remain unanswered related to the expression of specific surface markers of CD8+CD28− Tregs , the characterization of molecular pathways involved in generation and activation, the reason for their anergic status in vitro and the possible existence of modulators.
Many thanks for the comments. We agree that the manuscript should have included information on these areas. We have included the below at appropriate places within the text.
In contrast, STAT5 deficiency in the CD8+ T cell compartment is linked to impaired B cell tolerance due to an increase in CD8+Tfh resulting in increased autoantibody production [40]. These findings suggest TFs serve as tool to study CD8+ Treg populations and their differentiation [39,40].
The downregulation/loss of CD28 expression on the surface of CD8+ T cells coincide with normal aging [14]. Senescent CD8+CD28- T cells are oligoclonal, and express NK associated molecules CD56, CD57 and CD94 [81,82]. In cancer where senescent cells impact the immune response, further studies are warranted to examine the anergic, activation and exhausted phenotype compared to CD8+CD28-Tregs.
Overlapping markers between CD8+CD28-Tregs and CD4+CD25+Tregs include the expression of Foxp3, CD25 and members of the TNF receptor family (4-1BB) [49]. A unique feature mediated by CD8+CD28-Tregs is their ability to render APCs tolerogenic by reducing CD86 gene transcription [50] whilst upregulating inhibitory receptors: immunoglobulin like transcript (ILT) 3 (ILT3) and ILT4 [51]. Hence in the absence of CD28, this CD8+Treg population has adapted to respond to their microenvironment.
Few studies have characterized the molecular pathways involved in the generation and activation of CD8+CD28-Tregs. Mixed lymphocyte culture (MLC) stimulation and addition of IL-2, IL-7 and IL-15 have shown effective in generating and expanding CD8+Tregs [52].
Reviewer 2 Report
The topic of this review by Sabrina Ceeraz et al., is interesting and important due to the relevance of regulatory T cells in cancer biology and autoimmune/ autoinflammatory diseases. Needless to say, that Tregs function to limit the extent of tissue damage that might occur during a viral infection. However, not much is consistently known in literature about suppressor CD8 T cells. Comprehensive reviews are limiting and much of the information is scattered and focused on understanding the phenotype of CD8+ suppressor T cells. Although their presence and functions cannot be denied, the field still needs a lot of research in the years ahead.
Ceeraz et al., summarize the known phenotypes of CD8 suppressor cells, compare the literature on CD8 suppressors defined in mice vs humans and have then tried to compile the few papers which define a role for CD8 suppressor cells in autoinflammatory diseases like RA, SLE, SSc, MS and IBD etc.
Overall the review is good. However, I have one major concern. At several places either the correct original article has not been cited at all or a wrong citation has been added mistakenly. For a review article, references have to be those to original articles which unequivocally demonstrate the finding being discussed.
Please address the following major concerns regarding references:
- Line 57: Ref 12 does not talk about CD8 T cells in pregnancy but rather age-related effect on CD8 Tregs.
- Line 158: The study referring to work by Davila et al., 2005 is not cited and a reference #143 is mentioned which is non-existent.
- Line 160 and at other places: Authors say “we have shown” however they have not cited their work. Please cite the studies or mention “unpublished data or in work in progress”.
- Line 195: ref 37 is that on heart transplants and not SLE.
- Line 201: Ref 55 is a study on murine model of SLE and does not talk about human SLE. Please cite the right paper.
- Line 224: Regarding ref 53, cite the correct reference, or the original study.
It will be good to include the following two papers and refer to them in the paper.
1. CD8+ regulatory T cells are critical in prevention of autoimmune-mediated diabetes. Shimokawa et al., Nature communications 2020
2. CXCR5+PD-1+ follicular helper CD8 T cells control B cell tolerancehttps://www.nature.com/articles/s41467-019-12446-5.pdf
Minor comments/ Edits
- Replace “thanks to”, used in line 15 and 44 with “as a result of”.
- At all places please use CD8+ Tregs. Add a space between CD8+ and Tregs.
- Line 146: Change “on controls” to “in controlling”.
- Line 152: Expand MBP to Myelin basic protein.
- The authors are requested to go over the whole manuscript to make sure there are no r grammatical errors.
Author Response
Dear Reviewer,
Many thanks for your time in appraising the manuscript. Your comments have improved its scope and its readability, plus you identified a major error in referencing. We are very appreciative. Please find below our point-by-point answers to your comments.
Best wishes.
The topic of this review by Sabrina Ceeraz et al., is interesting and important due to the relevance of regulatory T cells in cancer biology and autoimmune/ autoinflammatory diseases. Needless to say, that Tregs function to limit the extent of tissue damage that might occur during a viral infection. However, not much is consistently known in literature about suppressor CD8 T cells. Comprehensive reviews are limiting and much of the information is scattered and focused on understanding the phenotype of CD8+ suppressor T cells. Although their presence and functions cannot be denied, the field still needs a lot of research in the years ahead.
Ceeraz et al., summarize the known phenotypes of CD8 suppressor cells, compare the literature on CD8 suppressors defined in mice vs humans and have then tried to compile the few papers which define a role for CD8 suppressor cells in autoinflammatory diseases like RA, SLE, SSc, MS and IBD etc.
Overall the review is good. However, I have one major concern. At several places either the correct original article has not been cited at all or a wrong citation has been added mistakenly. For a review article, references have to be those to original articles which unequivocally demonstrate the finding being discussed.
Please address the following major concerns regarding references:
- Line 57: Ref 12 does not talk about CD8 T cells in pregnancy but rather age-related effect on CD8 Tregs.
- Line 158: The study referring to work by Davila et al., 2005 is not cited and a reference #143 is mentioned which is non-existent.
- Line 160 and at other places: Authors say “we have shown” however they have not cited their work. Please cite the studies or mention “unpublished data or in work in progress”.
- Line 195: ref 37 is that on heart transplants and not SLE.
- Line 201: Ref 55 is a study on murine model of SLE and does not talk about human SLE. Please cite the right paper.
- Line 224: Regarding ref 53, cite the correct reference, or the original study.
Many apologies, this is extremely embarrassing. We had issues with an online Endnote library. These issues have been resolved. Many thanks for bringing the mistakes to our attention.
It will be good to include the following two papers and refer to them in the paper.
- CD8+regulatory T cells are critical in prevention of autoimmune-mediated diabetes. Shimokawa et al., Nature communications 2020
We thank the reviewer for the suggestion and have included the reference and a new paragraph on these cells in diabetes, as below.
Diabetes
In type 1 diabetic patients, treatment with an anti-CD3 mAb is shown to induce CD8+CD25+ Tregs which express Cytotoxic T-lymphocyte-associated protein (CTLA-4), and Foxp3 to facilitate inhibition of CD4+ T cells in response to therapy [65]. A recent study addressing how CD8+ Tregs may mediated suppression in diabetes reported that intestinal levels of Ruminococcus positively correlated with the number of CD8+ Tregs in the PB [66].
- CXCR5+PD-1+follicular helper CD8 T cells control B cell tolerancehttps://www.nature.com/articles/s41467-019-12446-5.pdf
Many thanks to the reviewer for the suggestion; we have included the reference in a two new sentences as outlined below.
In contrast, STAT5 deficiency in the CD8+ T cell compartment is linked to impaired B cell tolerance due to an increase in CD8+ Tfh resulting in increased autoantibody production [40]. These findings suggest TFs serve as tool to study CD8+ Treg populations and their differentiation [39,40].
We feel these two additional references greatly help the piece – many thanks.
Minor comments/ Edits
- Replace “thanks to”, used in line 15 and 44 with “as a result of”.
Many thanks; changed.
- At all places please use CD8+ Tregs. Add a space between CD8+ and Tregs.
Many thanks; all have been corrected.
- Line 146: Change “on controls” to “in controlling”.
Many thanks; changed.
- Line 152: Expand MBP to Myelin basic protein.
Many thanks for the suggestion, it has been changed. We have additionally added the below under the figure, to aid the reader.
Cytotoxic T-lymphocyte-associated protein 4 (CTLA-4); programmed death-1 (PD-1); T-cell immunoglobulin and mucin domain 3 (Tim-3); transforming growth factor beta (TGF-β).
- The authors are requested to go over the whole manuscript to make sure there are no r grammatical errors.
We have reviewed and found a couple of errors (e.g. human to humans), which we have corrected.
Round 2
Reviewer 1 Report
The manuscript has improved after addressing the raised points. However, due to potential therapeutic implications I would suggest to further expand the section "Viral Infections, Transplantation and Cancer". In particular, as the role of regulatory immune cell populations is currently of major interest in cancer treatment, I would suggest to mention the potential therapeutic implications of the emerging immuno-therapies targeting CD4+CD25 Regulatory FOXP3 Tcells, as recently described (Hepatocellular carcinoma in viral and autoimmune liver diseases: Role of CD4+ CD25+ Foxp3+ regulatory T cells in the immune microenvironment. World J Gastroenterol. 2021;27(22):2994-3009).
Author Response
Many thanks for this additional point. We agree that this section needed more depth and clarity. We have split the section and have included 2 additional paragraphs, plus the reference requested. Please see below. The section is highlighted in green in the manuscript. Many thanks for all your help in improving the piece. Best wishes, Richard
- Viral Infections
CD8+ Tregs have been documented in numerous viral infections. In individuals infected with hepatitis B virus (HBV), an increase in CD8+CD28- T cells in the chronic phase of infection negatively correlates with alanine aminotransaminase (ALT) and aspartate aminotransferase (AST) levels, suggesting improved liver health [76]. In human immunodeficiency virus (HIV) infected patients, an expansion in CD8+CD28- T cells has been reported in the blood and lung [77]. Similarly, these cells are additionally expanded in the blood of RA patients with Epstein-Barr Virus (EBV) infection [78].
- Transplantation
There is now increasing evidence that the interplay between co-inhibitory and co-stimulatory molecules play a role in CD8+CD28- Treg function. In transplantation studies, CD8+ T cells from B6 recipients of BALB/c heart allografts undergoing anti-ICOS Ab treatment prolong allograft survival, unlike CD8+ T cells isolated from control mice [18].
- Cancer
Increasing evidence points towards CD4+CD25+Foxp3+ Tregs having a pathogenic role in cancer development and metastasis, primarily through inhibiting host cancer-associated-antigen adaptive immunity [79,80]. As the numerically dominant Treg population in the tumour microenvironment (TME) CD4+CD25+Foxp3+ cells have been extensively studied in cancers, unlike CD8+CD28- Tregs. However, the few studies there are suggest similarities between the two populations. For example, an increase in CD8+CD28- Tregs has been seen to correlate with poor prognosis in metastatic cancer patients receiving T-lymphocyte immunotherapy following chemotherapy [81], and the expression of CD39 on CD8+Tregs has been linked to evasion of the antitumor immune response [82].
Inhibiting the impact of CD4+CD25+Foxp3+ Tregs through checkpoint blockade (CTLA4 and PD-L1;PD-1) has led to significant therapeutic breakthroughs in solid cancers [83]. Work is ongoing to try to deplete these cells, e.g., through CTLA-4 targeting, or inhibit their generation, e.g., inhibiting TGF-b1 signalling [80,84]. These concepts can equally be applied to CD8+CD28- Tregs, indeed, with CTLA-4 being reported on these cells [65], some of these current therapies may already be impacting this population.
The downregulation/loss of CD28 expression on the surface of CD8+ T cells has been shown to coincide with normal aging [14]. Senescent CD8+CD28- T cells are oligoclonal, and express NK associated molecules CD56, CD57 and CD94 [85,86]. In cancer, where senescent cells impact the immune response, further studies are warranted to examine the anergic, activation and exhausted phenotype compared to CD8+CD28- Tregs.
- Adoptive Cell Therapy.
Cell therapies such as engineered T cells expressing a specific chimeric antigen receptor (CAR) are efficacious as a non-solid cancer immunotherapy. A recent study where HLA-A*02 antigen-specific antibody (A2-CAR) human CD8+ Tregs reduces mismatched HLA-A*02 skin graft rejection and prevents xenogeneic GvHD in NSG mice [87] presents a novel platform for CD8+Treg therapy.
Reviewer 2 Report
The Authors have addressed my concerns.
Author Response
Many thanks again for all your help in improving the manuscript. All the very best, Richard.